# Exact Solutions and Degenerate Properties of Spin Chains with Reducible Hamiltonians

**Shiung Fan** 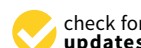

Beijing Computational Science Research Center, Beijing 100193, China; shiungfan@gmail.com

**Abstract:** The Jordan–Wigner transformation plays an important role in spin models. However, the non-locality of the transformation implies that a periodic chain of $N$ spins is not mapped to a periodic or an anti-periodic chain of lattice fermions. Since only the $N - 1$ bond is different, the effect is negligible for large systems, while it is significant for small systems. In this paper, it is interesting to find that a class of periodic spin chains can be exactly mapped to a periodic chain and an anti-periodic chain of lattice fermions without redundancy when the Jordan–Wigner transformation is implemented. For these systems, possible high degeneracy is found to appear in not only the ground state, but also the excitation states. Further, we take the one-dimensional compass model and a new XY-XY model ($\sigma_x \sigma_y - \sigma_x \sigma_y$) as examples to demonstrate our proposition. Except for the well-known one-dimensional compass model, we will see that in the XY-XY model, the degeneracy also grows exponentially with the number of sites.

**Keywords:** periodic spin chains; Jordan–Wigner transformation; degeneracy

## 1. Introduction

The Jordan–Wigner (JW) transformation establishes a connection between spin-1/2 operators and spinless fermion operators [1], and it has become a powerful tool for solving one-dimensional (1D) spin models and a few two-dimensional Ising models [2–4]. Besides, it is remarkable that the JW transformation has been generalized to higher dimensions in recent decades [5–13]. Typical examples of applications are provided in [2]. In that paper, using the JW transformation, Lieb et al. studied the ground states, excitations and the order of the one-dimensional XY model and Heisenberg–Ising model and concluded that both models have no long-range order for the isotropic case, but long-range order for any anisotropic cases.

Generally speaking, when the JW transformation is applied under the periodic boundary condition, theoretical physicists working on related fields always encounter that it introduces a phase term, and consequently causes redundant solutions. The work in [4] mentions this problem when the author introduces the JW transformation. As an example relevant to this paper, in [14], in order to exclude redundancy or to find the physical spectrum, Brzezicki et al. needed to distinguish the Bogoliubov vacuum by its parity and to judge whether operators change the parity. With regard to the Bogoliubov vacuum and fermion parity, [15] provided impressive clarification. The work in [15] not only introduced the basic concepts of the Bogoliubov quasiparticles and the Bogoliubov vacuum, but also rigorously discussed the choice of the Bogoliubov vacuum in general situations, the relationship between the particle-number parity and the Bogoliubov matrix transformation and the applications in systems owning the signature symmetry [16]. As for periodic spin chains, since the $N - 1$ bond takes an additional phase term $\exp(i\pi n)$ with $n = \sum_{l=1}^{N} a_l^\dagger a_l$ after the JW transformation, solutions depend on the evenness and oddness of the total number of occupancies, i.e., $n$, which is called the "$a$-cyclic" problem. To remove the redundancy, some effort is spent in doing projections;

or approximate results are adopted for large systems by directly dropping this phase, i.e., the "*c*-cyclic" problem [2].

　　In this paper, we find a class of systems in which the JW transformation does not introduce redundancy. In addition, it is discovered that the "holistic degeneracy" (or which can be interpreted as the degeneracy of the subspace of the Hamiltonian) exists in these systems, and it must be $2^x$-fold, in which $x$ is a positive integer. In some cases, the holistic degeneracy grows exponentially with the size of chains, and two representatives, the 1D compass model and a new XY-XY model, are given in Sections 3 and 4, respectively. Taking into consideration that a common feature of spin liquid states is the high degeneracy [17], the finding of this class of systems may help the research concerning spin liquid.

## 2. Spin-Fermion Mappings in Ordinary and Reducible Systems

　　For completeness, we first introduce how redundancy occurs in solutions, then give an abstract discussion on the solutions of reducible systems, and further, these systems are classified by the newly-defined holistic degeneracy.

　　Considering a 1D spin-1/2 system with $N$ sites labeled by $1, 2, \cdots, N$, its Hamiltonian is $H_s$. We assume that $H_s$ does not change the parity of the number of spin-up or -down states, and $H_s$ includes merely nearest-neighbor interactions. The restriction of nearest-neighbor interactions simplifies the problem, because merely the $N-1$ bond needs exceptional attention; otherwise, it becomes more complicated. Let the parity be $P_s = (-1)^{N_{up}}$ where $N_{up}$ is the number of spin-up states; we have:

$$[P_s, H_s] = 0. \tag{1}$$

　　Equation (1) indicates that the eigenstates of $H_s$ can be divided into two sets according to different eigenvalues of $P_s$. In one set, $P_s = 1$, and in another set, $P_s = -1$. Basic vectors in $H_s$'s Hilbert space $\mathcal{M}$ are described as:

$$\nu_i = \alpha_1 \otimes \alpha_2 \otimes \alpha_3 \cdots \otimes \alpha_N, i = 1, 2, 3, \ldots, 2^N, \tag{2}$$

where $\alpha$ is a spin-up or -down state. An eigenstate $\varphi$ of $H_s$ is the linear superposition of basic vectors, which can be expressed by:

$$\varphi = \sum_{i=1}^{2^N} \rho_i \nu_i, \tag{3}$$

where $\rho$ is the corresponding coefficient. Let $P_s$ act on $\varphi$; we have:

$$P_s \varphi = \sum_{i=1}^{2^N} \rho_i P_s \nu_i. \tag{4}$$

　　Since $P_s \varphi = \pm \varphi$, we have:

$$\sum_{i=1}^{2^N} \rho_i P_s \nu_i = \pm \sum_{i=1}^{2^N} \rho_i \nu_i. \tag{5}$$

　　Hence, $\nu$'s with nonzero $\rho$'s have the same parity with $\varphi$. For zero $\rho_i$, the parity of $\nu_i$ is not certain, yet this does not make sense. Accordingly, we divide $\mathcal{M}$ into two parts: $\mathcal{M} = \mathcal{M}_o \oplus \mathcal{M}_e$, in which $\mathcal{M}_o$ ($\mathcal{M}_e$) consists of basic vectors with odd-(even-)parity. By the assumptions, we have $H_s = H_s^o \oplus H_s^e$ (in matrix form), and $\mathcal{M}_o$ and $\mathcal{M}_e$ are the Hilbert spaces of $H_s^o$ and $H_s^e$, respectively. Obviously, $\mathcal{M}_o$ and $\mathcal{M}_e$ have equal dimensions $2^{N-1}$, i.e., $D(\mathcal{M}_o) = D(\mathcal{M}_e) = 2^{N-1}$. Independently diagonalizing $H_s^o$ and $H_s^e$, $2^{N-1}$ eigenvalues will be obtained for either one.

　　Now, we apply the JW transformation to $H_s$. The Pauli matrices in $H_s$ are transformed by the following relationships,

$$\sigma_l^x = \frac{\sigma_l^+ + \sigma_l^-}{2}, \sigma_l^y = \frac{\sigma_l^+ - \sigma_l^-}{2i}, \sigma_l^z = \frac{\sigma_l^+ \sigma_l^-}{4}, \tag{6}$$

where $\sigma^{\pm} = \sigma^x \pm i\sigma^y$, and the subscripts $l$ is the label of sites. The JW transformation is as follows:

$$\sigma_l^+ = 2a_l^\dagger \exp\left(i\pi \sum_{j<l} a_j^\dagger a_j\right),$$

$$\sigma_l^- = 2a_l \exp\left(-i\pi \sum_{j<l} a_j^\dagger a_j\right). \tag{7}$$

A one-to-one mapping between the spin-up (-down) state and the occupation (non-occupation) state of a fermion has been built, and meanwhile, the commutation relation of spin operators and anticommutation relation of fermion operators are preserved. We use the fermion operators to substitute for $\sigma^{\pm}$ in the spin Hamiltonian and obtain $H_f$. The purpose of implementing the JW transformation is to take advantage of the diagonalizable quadratic form of the fermion Hamiltonian $H_f$. A problem here that used to be faced on is the boundary condition. $H_s$ is considered to have the periodic boundary condition, i.e., $\sigma_N^{\pm}\sigma_{N+1}^{\pm} = \sigma_N^{\pm}\sigma_1^{\pm}$. Nevertheless, the equation $\sigma_N^{\pm}\sigma_{N+1}^{\pm} = \sigma_N^{\pm}\sigma_1^{\pm}$ may not be valid for $H_f$, because it depends on the parity of the number of occupation states. To clarify this statement, utilizing Equation (7), we have $\sigma_N^+\sigma_{N+1}^+ = 4a_N^\dagger a_{N+1}^\dagger$ and $\sigma_N^+\sigma_1^+ = -4\exp(i\pi n)a_N^\dagger a_1^\dagger$ with $n = \sum_{l=1}^N a_l^\dagger a_l$. Apparently for even $n$, $a_{N+1}^\dagger = -a_1^\dagger$, namely the anti-periodic boundary condition (APBC); for odd $n$, $a_{N+1}^\dagger = a_1^\dagger$, namely, the periodic boundary condition (PBC).

The same as $H_s$, $H_f$ does not change the parity of occupancies; we are able to divide the Hilbert space of $H_f$ into two subspaces: $\mathcal{M}^f = \mathcal{M}_o^f \oplus \mathcal{M}_e^f$. The dimensions of each space are:

$$D(\mathcal{M}^f) = 2^N, \ D(\mathcal{M}_o^f) = D(\mathcal{M}_e^f) = \frac{1}{2} \times 2^N. \tag{8}$$

To obtain exact results, we need to treat the Hamiltonian within the physical subspaces $\mathcal{M}_e^f$ and $\mathcal{M}_o^f$. Commonly, the dimensions of the Hamiltonian with a fixed boundary condition are two-times as large as the subspace, that is to say, $D(H_r^A) > D(\mathcal{M}_e^f)$ and $D(H_r^P) > D(\mathcal{M}_o^f)$, and consequently, redundancy is inevitable for solutions. In order to remove the redundancy, further projections are necessary.

Hereafter, the discussion turns to the contents we are focused on in this paper. We consider a class of systems in which the fermion Hamiltonians are reducible when the JW transformation is implemented. The reducible Hamiltonian means that the fermion Hamiltonian can be reduced to lower dimensions by appropriate methods, and such reductions always imply some symmetries in these systems. In Sections 3 and 4 and Appendices A and B, we give two examples to show how the dimensions of the Hamiltonians are reduced, and we will see that in both models, the Majorana fermion operators are of cruciality in the reduction. For these reducible systems, we denote the reduced Hamiltonian by $H_r$, and $H_r$ is represented by fermion operators. We further assume that $H_r$ has $Q$ quasiparticle states when it is diagonalized, which indicates that $D(H_r) = 2^Q$. $Q$ is definitely less than $N$. Let $H_r$ be limited to a fixed boundary condition, which explicitly does not change the dimensions of $H_r$, then we have $D(H_r^A) = 2^Q$ and $D(H_r^P) = 2^Q$. Defining $d = \frac{D(H_{s/f})}{D(H_r^A)+D(H_r^P)}$, we have two possible situations of $d$: 1. $d = 1$; 2. $d > 1$. When $d = 1$, we have $D(H_r^A) = D(\mathcal{M}_e^f)$ and $D(H_r^P) = D(\mathcal{M}_o^f)$; hence, $H_r$ can be exactly diagonalized in the physical subspaces, and redundancy is avoided. Therefore, $H_r$ with two boundary conditions exactly gives all solutions of $H_s$, and $N$ and $Q$ satisfy the following relation:

$$2^N = 2 \times 2^Q. \tag{9}$$

Otherwise, $d > 1$, and we find $D(H_r^A) < D(\mathcal{M}_e^f)$ and $D(H_r^P) < D(\mathcal{M}_o^f)$. Since $H_r$ is equivalent to $H_f$ except for the distinction of dimensions, it can be deduced that these systems own some kind of symmetries, such that a subspace can be divided into several smaller spaces that are all equivalent for $H_r$. Thereby, we have $H_f = dH_r^A \oplus dH_r^P$, where $d$ is referred as the multiplicity in the group theory. It deserves to be mentioned that the degeneracy of the ground state of $H_s$ is already determined simply

through the dimensions of $H_r$. Now that the subspace is divided, regarding each smaller space as an element, we are able to find quantum numbers like $q_1$, $q_2$, $q_3$, etc., and construct a complete set $\{H_r, q's\}$ to describe each smaller space. Basic vectors in each subspace are not always like these states defined in Equation (2), because the states in Equation (2) have been mixed by the Hamiltonian. Clearly, by deduction, the latter case $d > 1$ indicates that the degeneracy of all the eigenvalues given by $H_r$ is exactly $d$-fold (the degeneracy inside $H_r$ is not counted here). Since $d$ describes the degeneracy of a group of energy levels, it is more appropriate to call such degeneracy as holistic degeneracy. Similarly, for $d > 1$, we have:

$$2^N = 2d \times 2^Q. \tag{10}$$

Equation (9) can be regarded as a special case of Equation (10). From Equation (10), we easily find the relation between $d$, $N$, and $Q$,

$$d = 2^{N-Q-1}, \tag{11}$$

i.e., the holistic degeneracy increases exponentially with the degree of reduction. Further, it can be known that the total degeneracy for each energy level must be even-fold, and at least $d$-fold (here, the degeneracy within $H_r$ is taken into account).

At the end of this section, we conspicuously show our conclusions in Figure 1. In the upper mapping diagram of Figure 1a, it is seen that for ordinary systems, they are mapped to two half ranges belonging to the periodic and anti-periodic fermion chains, and each other half range is redundancy denoted by shadow areas. In contrast, in the lower mapping diagram of Figure 1b, reducible systems are mapped to full ranges of the periodic and anti-periodic fermion chains, and degeneracy exists.

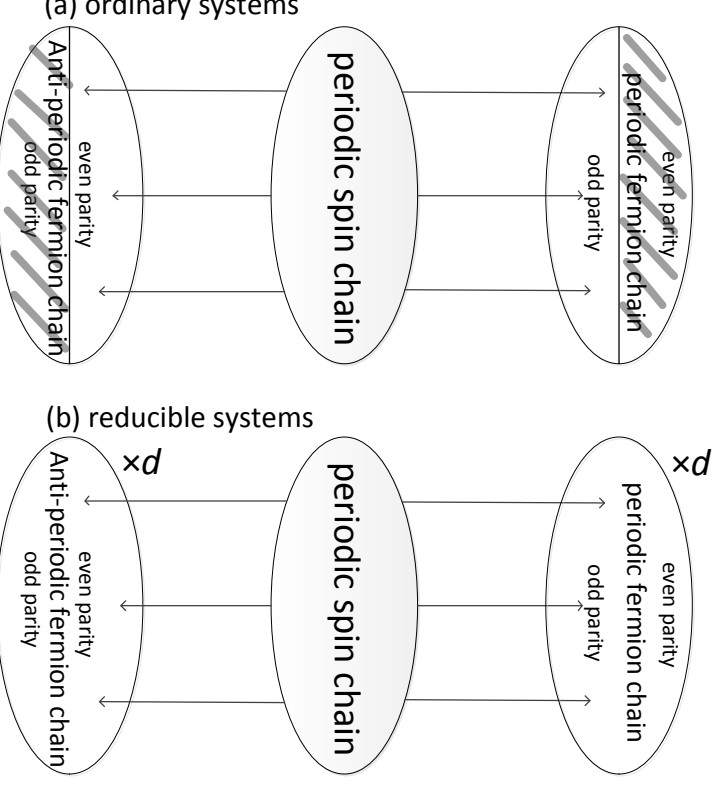

**Figure 1.** Mapping diagrams for (**a**) ordinary and (**b**) reducible systems. In (**a**), shadow areas mean the nonphysical states, i.e., the redundancy; in (**b**), marks "$\times d$" mean that the ranges are degenerate and the degeneracy is $d$-fold.

## 3. Example A: 1D Compass Model

In this part, we show an excellent example of our proposition. Generally speaking, in light of the degeneracy of these systems, it is in spin systems with special symmetries that our proposition is most possibly realized. At present, the compass model is known to own various symmetries [18]; hence, we naturally search a case in the compass model. Indeed, the 1D compass model [14,19] (for one's interest in recent progress, see [20–22]), which is also referred to as the reduced Kitaev model [23], is found to be a case of our proposition.

The 1D compass model has the following Hamiltonian:

$$H_s = J_x \sum_{l=1}^{N/2} \sigma_{2l-1}^x \sigma_{2l}^x + J_y \sum_{l=1}^{N/2} \sigma_{2l}^y \sigma_{2l+1}^y, \tag{12}$$

in which $J_x$ and $J_y$ are interacting parameters for odd and even bonds, respectively. We treat Equation (12) with the JW transformation and then substitute Majorana fermions for normal fermions. The Hamiltonian in the Majorana representation is:

$$H_{mf} = -i \sum_{l=1}^{N/2} \left( J_x c_{2l-1} c_{2l} - J_y c_{2l} c_{2l+1} \right), \tag{13}$$

and the diagonalization of Equation (13) has already been given in Appendix A. In Equation (13), the dimensions of the Hamiltonian are already reduced, because each site owns only one Majorana fermion with $\sqrt{2}$ degrees of freedom. Here, we stress that although the calculations in Appendix A are standard and the redundancy of ordinary systems can be discarded by carefully handling the parity of the states as mentioned in [2], our focus is on whether the JW transformation brings redundancy compared with conventional models.

Considering the PBC in Equation (12), since each Majorana fermion takes merely $\sqrt{2}$ degrees of freedom, $H_{mf}$ with a certain boundary condition has $Q = N/2$ quasiparticle states when it is diagonalized. Therefore, the dimensions of the Hamiltonian have been reduced from $2^N$ ($H_s$) to $2^{N/2}$ ($H_{mf}$). By the definition $d = \frac{D(H_s)}{D(H_{mf}^A) + D(H_{mf}^P)}$, we have two situations. If $d = 1$ (corresponding to the two-site case), $H_{mf}$ with the APBC and $H_{mf}$ with the PBC give eigenvalues in two physical subspaces, respectively, and they constitute all the eigenvalues of $H_s$ with no redundancy. Otherwise, $d > 1$, and under two boundary conditions, $H_{mf}$ gives a part of the eigenvalues of $H_s$, while no redundancy is introduced. Instead, the solutions are not complete. The approach of obtaining complete solutions is to duplicate the solutions of $H_{mf}^A$ and $H_{mf}^P$ for $d$ times. By Equation (11), we are able to find that $d = 2^{N/2-1}$. $d$ is called holistic degeneracy in this paper, and meanwhile, it represents the minimum degeneracy of the system. Therefore, it is straightforward to conclude that the ground state of the 1D compass model is $2^{N/2-1}$-fold degenerate, which is identical to the result obtained by the reflection positivity technique [24] and by mapping to the quantum Ising models [19].

To illustrate how the holistic degeneracy appears in the momentum space, utilizing the methods of [25], we elaborate the approach of finding all $2^N$ eigenvalues of the spin Hamiltonian in Appendix B.

Exploiting results in [14], we are able to analyze the symmetric characters in the real space. Rotating Equation (12) about the *x*-axis through $\pi/2$, i.e., $\sigma_y \to \sigma_z$, and $J_y \to J_z$, then the Hamiltonian in [14] is obtained,

$$H_s' = J_x \sum_{l=1}^{N/2} \sigma_{2l-1}^x \sigma_{2l}^x + J_z \sum_{l=1}^{N/2} \sigma_{2l}^z \sigma_{2l+1}^z. \tag{14}$$

Using the *z*-axis as the quantization axis, obviously the quantization axis is currently parallel with one interacting direction. Note that although the Hamiltonians of Equations (12) and (14) are equivalent, the methods in [14] are out of our formalism. By comparing the order of applying different

transformations between the methods of [14] and our formalism, one would find the difference. Transforming this Hamiltonian to the dual space by dividing the $N$-site chain into $N/2$ odd pairs, i.e., sites $2l-1$ and $2l$ constitute a unit. There are four states for each pair: $|\uparrow\uparrow\rangle, |\downarrow\downarrow\rangle, |\uparrow\downarrow\rangle, |\downarrow\uparrow\rangle$. Then, introducing a set of quantum numbers $\{s_1, s_2, \cdots, s_{N/2-1}, s_{N/2}\}$, $s_l$ corresponds to the $l$-th pair, and $s_l = 1$ for parallel states, while $s_l = 0$ for antiparallel states. Now, the Hilbert space can be divided equally into $2^{N/2}$ subspaces by giving the set with distinct values. The key point here is that the Hamiltonian in each subspace has the same solutions when $\sum_l s_l$ owns identical parity. Now, we think about symmetries in the dual space. First, for a certain $\sum_l s_l$, the set owns a permutation symmetry. For instance, when $\sum_l s_l = 1$, the Hamiltonians are the same wherever $s = 1$ is placed. Second, the Hamiltonians have no difference when $\sum_l s_l$ has the same parity. Thus, the condition that $\sum_l s_l$ is conserved modulo two for partial subspaces can be considered as a kind of symmetry here. To sum up, in the dual space, both the permutation and the modulo-two symmetries together result in the holistic degeneracy of the 1D compass model.

## 4. Example B: XY-XY Model

Except for the 1D compass model, it is easy to find another example of our proposition, which is as follows:

$$H = J \sum_{j=1}^{N} \sigma_j^x \sigma_{j+1}^y. \tag{15}$$

This model is named the XY-XY model here according to its form. The degenerate property is almost the same as that of the 1D compass model, except that it depends on the evenness and oddness of the number of sites. Besides, the method of solving the 1D compass model can be used on this Hamiltonian.

Define $c_j = i(a_j^\dagger - a_j)$. Applying the JW transformation, Equation (15) has the form:

$$H = iJ \sum_{j=1}^{N} c_j c_{j+1}. \tag{16}$$

Then, applying the Fourier transformation, we obtain:

$$H = 2J \sum_k \sin k\, c_k^\dagger c_k, \tag{17}$$

where $c_k^\dagger = \frac{1}{\sqrt{2N}} \sum_j \exp(ikj) c_j$, and $k$ has the values in Equation (A12), but with $N' = N$. Besides, we have the constraints $c_k^\dagger c_k + c_{-k}^\dagger c_{-k} = 1$ and $c_\pi^\dagger c_\pi + c_0^\dagger c_0 = 1$. The spectrum of the XY-XY model is gapless and identical to the $J_x = J_y$ case of the 1D compass model. However, a subtle difference exists. The number of sites is even for the 1D compass model; in contrast, that can be even or odd for the XY-XY model. For even $N$, the holistic degeneracy is $d = 2^{N/2-1}$. For odd $N$, the $k = \pi/k = 0$ state under the APBC/PBC can be directly eliminated since $\sin k = 0$; and the spectra under the APBC and PBC are the same according to the trigonometric function $\sin k_{APBC} = \sin(\pi - k_{APBC}) = \sin k'_{PBC}$, which means that the state $k_{APBC}$ corresponds to a state $k'_{PBC}$. Therefore, for odd $N$, the holistic degeneracy is $d = 2^{(N+1)/2}$. According to Appendix B, the same results can be obtained using normal fermion operators.

For the purpose of analyzing the symmetry, like the 1D compass model, the XY-XY chain can be mapped to the quantum Ising model [2,19,26]. We utilize the eigenstates of $\sigma^y$ and $\sigma^x$ to represent states in sites $2l-1$ and $2l$, respectively. For example, $|Y(X)_1\rangle$ and $|Y(X)_{-1}\rangle$ represent spin-up and -down states in the $y(x)$-axis. For the odd pairs, i.e., bonds $(2l-1)$–$2l$, we label the states of $|Y_1, X_1\rangle$

and $|Y_{-1}, X_{-1}\rangle$ with $s_l = 1$, and other states with $s_l = 0$. Then, define pseudospin operators for odd pairs with $s = 1$,

$$
\begin{aligned}
\Gamma_l^x &= |Y_1, X_1\rangle\langle Y_{-1}, X_{-1}| + |Y_{-1}, X_{-1}\rangle\langle Y_1, X_1|, \\
\Gamma_l^z &= |Y_1, X_1\rangle\langle Y_1, X_1| - |Y_{-1}, X_{-1}\rangle\langle Y_{-1}, X_{-1}|.
\end{aligned}
\tag{18}
$$

By analogy, for odd pairs with $s = 0$,

$$
\begin{aligned}
\Gamma_l^x &= |Y_1, X_{-1}\rangle\langle Y_{-1}, X_1| + |Y_{-1}, X_1\rangle\langle Y_1, X_{-1}|, \\
\Gamma_l^z &= |Y_1, X_{-1}\rangle\langle Y_1, X_{-1}| - |Y_{-1}, X_1\rangle\langle Y_{-1}, X_1|.
\end{aligned}
\tag{19}
$$

As is seen in Figure 2, we divide all sites into odd pairs; however, in Figure 2b, the end site $N$ is isolated when $N$ is odd. Its operators can be individually defined,

$$
\begin{aligned}
\Gamma_{\frac{N-1}{2}+1}^x &= |Y_1\rangle\langle Y_{-1}| + |Y_{-1}\rangle\langle Y_1|, \\
\Gamma_{\frac{N-1}{2}+1}^z &= |Y_1\rangle\langle Y_1| - |Y_{-1}\rangle\langle Y_{-1}|.
\end{aligned}
\tag{20}
$$

Since the isolated site $N$ already has two degrees of freedom, we do not assign the label $s$ to it. Now, each subspace can be labeled by a set $\{s_1, ..., s_{N/2\,or\,(N-1)/2}\}$.

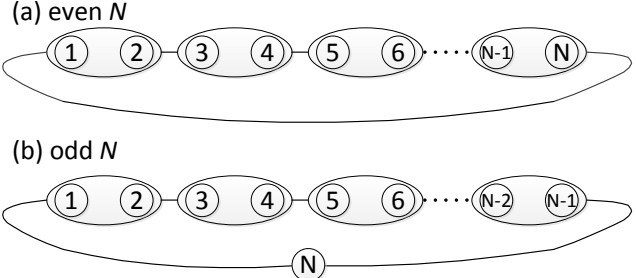

**Figure 2.** Depictions of the XY-XY model with (**a**) an even number of sites and (**b**) an odd number of sites. Circles around every two sites denote the odd pairs in the text. When the number of sites is odd, the end site $N$ does not form pairs.

By our definitions, the Hamiltonian can be transformed to:

$$
H_{even} = \sum_{l=1}^{N/2} (-1)^{s_l} \Gamma_l^x + (-1)^{s_l+1} \Gamma_l^z \Gamma_{l+1}^z,
$$

$$
H_{odd} = \sum_{l=1}^{(N-1)/2-1} (-1)^{s_l} \Gamma_l^x + \sum_{l=1}^{(N-1)/2-1} (-1)^{s_l+1} \Gamma_l^z \Gamma_{l+1}^z + \Gamma_{(N-1)/2+1}^x \Gamma_1^z,
\tag{21}
$$

where the subscripts "even" and "odd" denote the parity of the number of sites. Except for the boundary term, other sign factors can be removed by canonical transformations, as a result,

$$
H_{even} = \sum_{l=1}^{N/2} \Gamma_l^x + \sum_{l=1}^{N/2-1} \Gamma_l^z \Gamma_{l+1}^z + (-1)^{N/2+2s_{N/2}-\sum s} \Gamma_{N/2}^z \Gamma_1^z,
$$

$$
H_{odd} = \sum_{l=1}^{(N-1)/2-1} \Gamma_l^x + \Gamma_l^z \Gamma_{l+1}^z + \Gamma_{(N-1)/2}^z \Gamma_{(N-1)/2+1}^z + \Gamma_{(N-1)/2+1}^x \Gamma_1^z.
\tag{22}
$$

Then, $H_{even}$ is exactly solvable, and the calculations can be found in [14]. Nevertheless, when we apply the JW transformation, the quadratic form of $H_{odd}$ is not accessible because of the $\Gamma^x\Gamma^z$ term. Hence, in this situation, our approach with the help of Majorana fermions shows its priority.

In addition, it is noticed that the boundary term of $H_{odd}$ does not take a sign factor, and at present $d = 2^{(N-1)/2}$, according to the previous result $d = 2^{(N+1)/2}$, each subspace must own extra two-fold holistic degeneracy. It is not hard to figure out the absent holistic degeneracy. To construct new basic vectors that reduce the size of the subspace, we need to label each basic vector. When $\sigma_x^{2l}\sigma_y^{2l+1}|X_{2l}, Y_{2l+1}\rangle = 1 \ (-1)$, we label the corresponding basic vectors with $t_l = 1 \ (-1)$, then each basic vector of a specific subspace is labeled by a set $\{n_1, ..., n_{(N-1)/2}\}$. It is straightforward to realize that two basic vectors share a common set. We now construct new basic vectors by linearly combining the two old vectors with the same label,

$$
\begin{aligned}
v_{even}^{new} &= v_1^{old} + v_2^{old}, \\
v_{odd}^{new} &= v_1^{old} - v_2^{old},
\end{aligned}
\tag{23}
$$

where the subscripts "even" and "odd" denote the parity of new basic vectors. Letting $H_{odd}$ act on new vectors, it is found that the subspace is divided by the parity of new vectors, and the Hamiltonian matrices are the same. We take the three-site system as an example to display the effectiveness of our approach. One subspace of the three-site model includes the following basic vectors:

$$
|Y_1, X_1, Y_1\rangle, |Y_1, X_1, Y_{-1}\rangle, |Y_{-1}, X_{-1}, Y_1\rangle, |Y_{-1}, X_{-1}, Y_{-1}\rangle.
\tag{24}
$$

A new smaller subspace is made up of:

$$
|Y_1, X_1, Y_1\rangle + |Y_{-1}, X_{-1}, Y_{-1}\rangle, |Y_{-1}, X_{-1}, Y_1\rangle + |Y_1, X_1, Y_{-1}\rangle.
\tag{25}
$$

Obviously, $H_{odd}$ does not mix both vectors with outside vectors. Finally, the absent holistic degeneracy is located.

In a word, similar to the 1D compass model, the holistic degeneracy of the XY-XY model mostly comes from the different configurations of odd pairs or the parity of $s_l$. In particular, when $N$ is odd, two-fold holistic degeneracy is from the parity of new basic vectors.

## 5. Conclusions

To sum up, we have theoretically proposed that, with no redundancy, the JW transformation can exactly map a periodic spin chain to a periodic, and an anti-periodic chain of lattice fermions when the Hamiltonians in the fermion representation can be reduced to lower dimensions. The conditions include that the Hamiltonian merely involves the nearest-neighbor interactions and does not change the parity of the number of fermions. In these systems, the holistic degeneracy is defined, and it is found to be $2^x$-fold where $x$ is a positive integer. These systems are further classified according to the folds of holistic degeneracy, and possible high degeneracy exists in these systems. In addition, we take the 1D compass model and the XY-XY model as the examples to demonstrate the degenerate properties of these systems. In both models, their holistic degeneracy grows exponentially with the size of systems. It is remarkable that by our work, complete energy spectra can be totally determined by the reduced fermion Hamiltonian with little effort, although other advantages are not clear yet.

**Funding:** This research received no external funding.

**Acknowledgments:** S.F. thanks Hai-Qing Lin and Jian Lee for useful discussions. S.F. acknowledges the support from Beijing Computational Science Research Center and  Hai-Qing Lin.

**Conflicts of Interest:** The author declares no conflict of interest.

## Appendix A. Diagonalization of Equation (13)

The Hamiltonian for the 1D compass model is:

$$H = J_x \sum_{l=1}^{N/2} \sigma_{2l-1}^x \sigma_{2l}^x + J_y \sum_{l=1}^{N/2} \sigma_{2l}^y \sigma_{2l+1}^y, \tag{A1}$$

where $J_x$ is the interacting strength for the odd bonds and $J_y$ is for the other half of bonds. A toy model for this Hamiltonian is depicted in Figure A1a. We use the JW transformation to transform the spin Hamiltonian and have:

$$\begin{aligned} H = & J_x \sum_{l=1}^{N/2} (a_{2l-1}^\dagger a_{2l}^\dagger + a_{2l-1}^\dagger a_{2l} + H.c.) \\ & + J_y \sum_{l=1}^{N/2} (-a_{2l}^\dagger a_{2l+1}^\dagger + a_{2l}^\dagger a_{2l+1} + H.c.). \end{aligned} \tag{A2}$$

Taking advantage of the Majorana fermion operators, the Hamiltonian has a more concise form:

$$H = -i \sum_{l=1}^{N/2} (J_x c_{2l-1} c_{2l} - J_y c_{2l} c_{2l+1}), \tag{A3}$$

where $c_{2l-1} = i(a_{2l-1}^\dagger - a_{2l-1})$ and $c_{2l} = a_{2l}^\dagger + a_{2l}$ are Majorana fermion operators. The corresponding illustration is given in Figure A1b. It is known that a normal fermion can be described by a pair of Majorana fermions, and accordingly, each Majorana fermion has $\sqrt{2}$ degrees of freedom. Interestingly, the Hamiltonian here does not include the other half Majorana fermions d's which pair with c's. Through counting the number of the Majorana fermions, it is found that the dimensions of the Hamiltonian have been reduced compared with the original spin Hamiltonian.

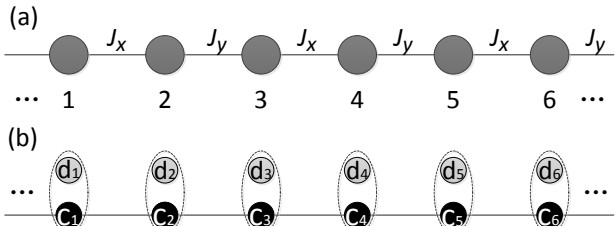

**Figure A1.** (**a**) A toy model for the 1D compass model and (**b**) the Majorana representation of this model. $J_x$ and $J_y$ are the interacting strength between two nearest sites, and $J_x$ and $J_y$ are for the $x$-axis and $y$-axis interactions, respectively. c's and d's are paired Majorana states; remarkably, c's are exclusively involved in the Majorana representation.

To diagonalize the Hamiltonian, we mark odd sites with A's, and even sites B's. Thus, $N$ sites are divided into $N/2$ cells. Equation (A3) becomes:

$$\begin{aligned} H = & -i \sum_{l=1}^{N/2} (J_x c_{l,A} c_{l,B} - J_y c_{l,B} c_{l+1,A}) \\ = & -\frac{i}{2} \sum_{l=1}^{N/2} (J_x c_{l,A} c_{l,B} - J_x c_{l,B} c_{l,A} - J_y c_{l,B} c_{l+1,A} \\ & + J_y c_{l+1,A} c_{l,B}), \end{aligned} \tag{A4}$$

where $l$ is the label of cells. Equation (A4) can be described by:

$$\Psi^\dagger H_{ij}\Psi, \tag{A5}$$

where $\Psi^\dagger = \begin{pmatrix} \cdots & c_{jA} & c_{jB} & \cdots \end{pmatrix}$, $\Psi = \begin{pmatrix} \vdots \\ c_{jA} \\ c_{jB} \\ \vdots \end{pmatrix}$, and $H_{ij}$ is a skew symmetric matrix. Further,

transform Equation (A4) to the momentum space by the Fourier transformation:

$$\Phi^\dagger = \Psi^\dagger \frac{1}{\sqrt{2}} F^\dagger,$$

$$\Phi = \frac{1}{\sqrt{2}} F\Psi, \tag{A6}$$

where $\Phi^\dagger = \begin{pmatrix} \cdots & c_{kA}^\dagger & c_{kB}^\dagger & \cdots \end{pmatrix}$, $\Phi = \begin{pmatrix} \vdots \\ c_{kA} \\ c_{kB} \\ \vdots \end{pmatrix}$ and $F^\dagger/F$ is the matrix of a normal Fourier

transformation from the real space to the momentum space. The purpose of multiplying a factor $\frac{1}{\sqrt{2}}$ before $F^\dagger/F$ is to make the operators in the momentum space satisfy the anti-commutation relation. $c_{k,A/B}^\dagger$ and $c_{k,A/B}$ are as follows:

$$c_{k,A/B}^\dagger = \frac{1}{\sqrt{N}} \sum_j \exp\left(ikR_j\right)c_{j,A/B},$$

$$c_{k,A/B} = \frac{1}{\sqrt{N}} \sum_j \exp\left(-ikR_j\right)c_{j,A/B}. \tag{A7}$$

It can be verified that $c_{k,A/B}^\dagger = c_{-k,A/B}$ and $\{c_{kv}^\dagger, c_{k'v'}\} = \delta_{kk'}\delta_{vv'}$ ($v = A, B$). In the momentum space, the Hamiltonian becomes:

$$H = \Phi^\dagger F 2 H_{ij} F^\dagger \Phi, \tag{A8}$$

which can be divided into blocks. Each block is written as:

$$H_k = \begin{pmatrix} c_{kA}^\dagger & c_{kB}^\dagger \end{pmatrix} \begin{pmatrix} 0 & f(k) \\ f(k)^* & 0 \end{pmatrix} \begin{pmatrix} c_{kA} \\ c_{kB} \end{pmatrix} \tag{A9}$$

where $f(k) = -iJ_x - iJ_y \exp\left(-ik\right)$. Then, exploiting the Bogoliubov transformation to diagonalize $H_k$, we obtain:

$$H_k = |f(k)|\alpha_k^\dagger \alpha_k - |f(k)|\beta_k^\dagger \beta_k, \tag{A10}$$

where $\alpha_k^\dagger = \frac{f(k)}{\sqrt{2}|f(k)|}c_{kA}^\dagger + \frac{1}{\sqrt{2}}c_{kB}^\dagger$ and $\beta_k^\dagger = \frac{f(k)}{\sqrt{2}|f(k)|}c_{kA}^\dagger - \frac{1}{\sqrt{2}}c_{kB}^\dagger$. Meanwhile, considering the relation $c_{k,A/B}^\dagger = c_{-k,A/B}$, we have $\alpha_k^\dagger \alpha_k + \beta_{-k}^\dagger \beta_{-k} = 1$ and $\beta_k^\dagger \beta_k + \alpha_{-k}^\dagger \alpha_{-k} = 1$. In particular, when $k = 0$ or $\pi$, we have $\alpha_k^\dagger \alpha_k + \beta_k^\dagger \beta_k = 1$. These constraints are radically attributed to the Majorana fermions' own symmetry, and each constraint cut off a half of solutions of the Hamiltonian. With the constraints, the diagonal Hamiltonian is:

$$H = \sum_k 2E_k(\alpha_k^\dagger \alpha_k - \frac{1}{2}), \tag{A11}$$

where $E_k = |f(k)| = \sqrt{J_x^2 + J_y^2 + 2\cos k J_x J_y}$ is the energy of $k$-mode quasiparticle states. Under the adopted form, $-E_k$ corresponds to $-k$-mode quasiparticle states, except for $k = 0$ and $\pi$.

Till Equation (A11), although we transform the Hamiltonian under the periodic condition $\sigma_{N+1} = \sigma_1$, the boundary conditions for the fermion Hamiltonian are not fixed yet. Now, we assign values to $k$ to identify the boundary conditions. For the APBC, the momentum $k$ has values:

$$\pm \frac{1}{N'}\pi, \pm\frac{3}{N'}\pi, \pm\frac{5}{N'}\pi, \cdots, \pm\frac{N'-1}{N'}\pi \,(\text{even } N'), \pi \,(\text{odd } N');$$

for the PBC, $k$ has values:

$$0, \pm\frac{2}{N'}\pi, \pm\frac{4}{N'}\pi, \cdots, \pm\frac{N'-1}{N'}\pi \,(\text{odd } N'), \pi \,(\text{even } N'), \tag{A12}$$

where $N' = N/2$. For each boundary condition, there must be $N'$ quasiparticle states, and Equation (A11) will give $2^{N'}$ energy levels for a $N$-site system.

## Appendix B. Proof of the Holistic Degeneracy in the 1D Compass Model

Except for the discussion on degeneracy, the following method is from [25]. Similar to Appendix A, the odd site is marked with A, and the even site B. Equation (A2) becomes:

$$
\begin{aligned}
H = & J_x \sum_{l=1}^{N/2} (a_{l,A}^\dagger a_{l,B}^\dagger + a_{l,A}^\dagger a_{l,B} + H.c.) \\
& + J_y \sum_{l=1}^{N/2} (-a_{l,B}^\dagger a_{l+1,A}^\dagger + a_{l,B}^\dagger a_{l+1,A} + H.c.).
\end{aligned}
\tag{A13}
$$

Then, using the Fourier transformation $c_{l,A(B)}^\dagger = \frac{1}{\sqrt{N/2}}\sum_k \exp(-ikR_l){c_{k,A/B}}^\dagger$, Equation (A13) is directly transformed to the momentum space:

$$
\begin{aligned}
H = & J_x \sum_k (a_{k,A}^\dagger a_{-k,B}^\dagger + a_{k,A}^\dagger a_{k,B} + H.c.) \\
& + J_y \sum_k (-\exp(ik)a_{k,B}^\dagger a_{-k,A}^\dagger + \exp(ik)a_{k,B}^\dagger a_{k,A} + H.c.),
\end{aligned}
\tag{A14}
$$

in which $-k$ should be modified to $k$ when $k = \pi$. Define:

$$
\begin{aligned}
H_k = & J_x(a_{k,A}^\dagger a_{-k,B}^\dagger + a_{k,A}^\dagger a_{k,B} + H.c.) \\
& + J_y(-\exp(ik)a_{k,B}^\dagger a_{-k,A}^\dagger + \exp(ik)a_{k,B}^\dagger a_{k,A} + H.c.);
\end{aligned}
\tag{A15}
$$

and:

$$W(k) = H_k + H_{-k} \,(0 < k < \pi). \tag{A16}$$

Since the Hamiltonian has already been decoupled in the $k$ representation, we have:

$$H = H_0 + H_\pi + \sum_{0 < k < \pi} W(k), \tag{A17}$$

and each part can be solved independently. For each $W(k)$, its Hilbert space has sixteen dimensions, and the Hilbert space can be divided into two subspaces with the same dimensions.

Firstly, we solve $W(k)$ in the subspace with even parity. The subspace with even parity has the following basic vectors:

$$|0\rangle, \quad a_{k,A}^{\dagger} a_{-k,A}^{\dagger} |0\rangle, \quad a_{k,B}^{\dagger} a_{-k,B}^{\dagger} |0\rangle,$$
$$a_{k,A}^{\dagger} a_{k,B}^{\dagger} |0\rangle, \quad a_{k,A}^{\dagger} a_{-k,B}^{\dagger} |0\rangle, \quad a_{-k,A}^{\dagger} a_{k,B}^{\dagger} |0\rangle,$$
$$a_{-k,A}^{\dagger} a_{-k,B}^{\dagger} |0\rangle, \quad a_{k,A}^{\dagger} a_{-k,A}^{\dagger} a_{k,B}^{\dagger} a_{-k,B}^{\dagger} |0\rangle. \tag{A18}$$

The eigenvalues of $W(k)$ in this subspace are:

$$\lambda_1^e = 2E(k), \lambda_2^e = -2E(k), \lambda_3^e = 0, \lambda_4^e = 0,$$
$$\lambda_5^e = 2E(k), \lambda_6^e = -2E(k), \lambda_7^e = 0, \lambda_8^e = 0, \tag{A19}$$

where $E(k)$ is identical to that in Appendix A. In the subspace with odd parity, $W(k)$ has eigenvalues:

$$\lambda_1^o = 2E(k), \lambda_2^o = -2E(k), \lambda_3^o = 0, \lambda_4^o = 0,$$
$$\lambda_5^o = 2E(k), \lambda_6^o = -2E(k), \lambda_7^o = 0, \lambda_8^o = 0. \tag{A20}$$

One needs to be careful with the point that it is the subspace of $W(k)$ rather than $H$. Therefore, Equations (A19) and (A22) are valid for both boundary conditions. Similarly, for $H_0$,

$$\lambda_1^e = E(0), \lambda_2^e = -E(0),$$
$$\lambda_1^o = E(0), \lambda_2^o = -E(0); \tag{A21}$$

for $H_\pi$,

$$\lambda_1^e = E(\pi), \lambda_2^e = -E(\pi),$$
$$\lambda_1^o = E(\pi), \lambda_2^o = -E(\pi). \tag{A22}$$

Whether the terms $H_0$ and $H_\pi$ exist in the Hamiltonian depends on the boundary conditions (see Equation (A12) in Appendix A).

Now, we show how to find all eigenvalues of the 1D compass model. Defining a set of quantum numbers $\{q_0, \cdots, q_k, \cdots, q_\pi\}$, we let $q_k = 1$ when selecting an eigenvalue corresponding to odd parity ($\lambda^o$) from $W(k)$ and let $q_k = 0$ when selecting a $\lambda^e$. For the case of the APBC and even $N/2$, we need to satisfy the condition of the even-parity eigenstate; hence, we must have:

$$\mathrm{mod}\ (\sum_k q_k, 2) = 0. \tag{A23}$$

There are $N/4$ $q$'s in the set $\{q_k\}$ ($0 \leq k \leq \pi$); thus, $2^{N/4-1}$ out of $2^{N/4}$ cases are appropriate, like the case $\sum_k q_k = 0$, which means that always choosing a $\lambda^e$ from each $W(k)$. Besides, it is also noticed that each eigenvalue of $W(k)$ in a certain subspace has two-fold holistic degeneracy; therefore, we obtain $2^{N/4}$-fold holistic degeneracy for each appropriate case of $\{q_k\}$. Incorporating both factors leads to the $2^{N/2-1}$-fold holistic degeneracy. The analysis above can be extended to other situations straightforwardly, and the conclusions are consistent.

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
