# Peer review of "Exact Solutions and Degenerate Properties of Spin Chains with Reducible Hamiltonians"

_condensedmatter, doi:10.3390/condmat3040032_

Round 1
Reviewer 1 Report
The present work uses a mapping to give a solution to the spectra of a reduced fermion Hamiltonian usually applied to problems of spin chains.
The manuscript uses expression like ''amazing'', etc. This should be avoided. I suggest the author to rephrase the scientific language of the paper in a more sober style. The introduction is considerably technical, neither is clear why a generic reader of should be interested in this problem. A paragraph should be included giving some examples of condensed matter physical problems where this calculations could find some application or some interest.
The author use sentence like: "Normally, when the JW transformation
is implemented under the periodic boundary condition, people always encounter that it introduces
redundant solutions to the quadratic Hamiltonians" that are too colloquial and also imply that the reader is a real expert of the topic that is discussed. The author doesn't even bother to put the references to the work of these "people".
The author should spend some more words in the introduction to explain what is "a-cyclic" and "c-cyclic" problems without assuming previous knowledge from the reader or by just adding the reference. The author also should explain what is the advantage to have found for a class of systems solution to the JW transformation that does not include redundancy. Declaring that special degenerate properties are discussed in the paper, really does not add much information in the introduction, it should be better explain what the author mean.
The paragraph named: Law, it is really generic and it is not clear what information should give to the reader. The title of the paragraph should be improved and made more clear. The author should make a paragraph dedicated especially to the mapping since it is essential for the calculations and is one of the most interesting point of the paper. Again, the paragraph with the title examples it is really not clear. The author should give a title for each one of the examples reported.
The paper need to be improved since it assumes that all the readers of "condensed matter" are specialist in the field of JW transformations. Introduction and discussion of the results need to be improved. I can't recommend the publication of this paper in the present form.
Reviewer 2 Report
-------------------------------------------------------
Referee's Report on condensedmatter-365277/Fan
-------------------------------------------------------
The manuscript reports high degeneracy of exactly obtained
ground and excited states of frustrated spin chains. This
problem is of current interest but several properties were
already discussed in the past. The model proposed in Eq. (16)
is new and the discussion should concentrate around it. The
1D compass model has been discussed extensively in the past
and this discussion has to be reduced before the paper could
be reconsidered.
At present, the following issues are discussed in an
incomplete way and presentation should be improved there:
(1) The degeneracy of the ground state in the 1D compass
model is well known and amounts to $2^{N/2-1}$. Altogether
the problem is trivial knowing the discussion which followed.
Reference [15] has a misprint here and these authors confirm
the same number of degenerate states in the ground state in:
W. Brzezicki and A. M. Ole\'s, Acta Phys. Polon. A 115, 162
(2009). This paper should be cited next to [15] and [19] to
avoid any confusion and to highlight the degeneracy.
(2) The last sentence of the abstract is not a new result
and should be removed. Anyway, the present study has its own
standing and this impression should be made instead of citing
previous papers in the abstract and providing the misleading
motivation. The remaining content of the abstact reflects
already a few new results, but also the model Eq. (16) could
be mentioned at the end of the abstract.
(3) A detailed discussion about fermion parity in the vacuum
state of Bogoliubov quasiparticles is given in:
George Bertsch, Jacek Dobaczewski, Witold Nazarewicz, and
Junchen Pei, Phys. Rev. A 79, 043602 (2009).
This paper should be cited and the advance made by the Author
should be discussed in this context.
(4) A better explanation of the role played by Majorana
fermions would be very helpful. After the transformation (A8)
only conventional fermions are used, so the advantage
of using Majorana fermions is unclear.
(5) The sentence "Details are left for discussions behind."
below Eq. (4) seems to be misplaced as there are no more
"details" to which this sentence refers. It is advised that
this sentence is simply removed.
(6) Discussion of the boundary conditions for the spin
Hamiltonian below Eq. (A4) is obvious and it is clear that
the problem has to be solved in subspaces and the discussion
of this property is obsolete.
In conclusion, a properly revised manuscript could be
reconsidered for publication in Condensed Matter.
Publication of the present manuscript cannot be recommended.
Reviewer 3 Report
The author addresses a specific question about the effectiveness of the Jordan Wigner trasformation in spin sistems. In my opinion the proposed results are correct, but of limited interest and relevance. In any case I think that the paper deserves to be published since it can be useful for the researchers working on Jordan Wigner trasformation.
Round 2
Reviewer 1 Report
I recommend the paper for publication.
Reviewer 2 Report
Second Refereee's Report onManuscript IDcondensedmatter-365277
====================================================
The revised manuscript is more balanced and emphasizes new results.
The Author followed the advice in the first report and made appropriate
changes. More references were included.
It is my pleasure to recommend the revised manuscript for publication.